# Note on Common Fixed Point Theorems in Convex Metric Spaces

**Anil Kumar** [1,†] and **Aysegul Tas** [2,*,†]

1 Department of Mathematics, Government College for Women Sampla, Rohtak 124501, Haryana, India; anilkshk84@gmail.com
2 Department of Management, Cankaya University, 06790 Ankara, Turkey
* Correspondence: aysegul@cankaya.edu.tr
† These authors contributed equally to this work.

**Abstract:** In the present paper, we pointed out that there is a gap in the proof of the main result of Rouzkard et al. (The Bulletin of the Belgian Mathematical Society 2012). Then after, utilizing the concept of (E.A.) property in convex metric space, we obtained an alternative and correct version of this result. Finally, it is clarified that in the theory of common fixed point, the notion of (E.A.) property in the set up of convex metric space develops some new dimensions in comparison to the hypothesis that a range set of one map is contained in the range set of another map.

**Keywords:** compatible maps; common fixed points; convex metric spaces; q-starshaped

## 1. Introduction

A point on which a self-map remains invariant is called a fixed point for that map. Fixed point theory plays an important role in solving different kinds of problems of nonlinear analysis and so it has applications in engineering, medical science, physical science, computer science, etc. In 1922, Banach [1] proved that a contraction map in a complete metric space has a fixed point and this result is known as the Banach contraction principle. Due to the simplicity and usefulness of this result, fixed point theory became a more aggressive area of research. Many researchers so far have worked in this field, extending this contraction principle in several possible ways [2–4].

In 1976, Jungck [5] extended the Banach contraction principle for the pair of commuting self-maps by ensuring the existence of a common fixed point for this pair. Sessa [6] relaxed the condition of commutativity and introduced the class of weak commuting maps. Again, Jungck [7] gave the weaker version of the commutativity condition by introducing the class of compatible maps and proved that weak commuting maps are compatible but the converse is not true in general. After that, many authors obtained more comprehensive common fixed point theorems under some given hypothesis [8,9].

On the other side, Takahashi [10] defined the notion of a convex structure in a metric space and called such a space a convex metric space. Further, he studied several properties of this space and ensure the existence of a fixed point for nonexpansive maps in the setup of convex metric space. In the last forty years, many fixed point and common fixed point theorems in the context of convex metric space have been established; for example, see [11–15].

## 2. Preliminaries

In the present section we recall some standard notations, basic definitions and auxiliary results, which are required in the sequel.

In 2014, inspired by the idea of Aamri and Moutawakil [16], the concept of (E.A.) property in the context of convex metric space was introduced by Kumar and Rathee [17].

In the present article, we shall show that there is a gap in the proof of the Theorem 1, which is one of the main results of Rouzkard et al. [18]. Then, we obtain a correct version

of Theorem 1 by utilizing the concept of (E.A.) property in a convex metric space defined by Kumar and Rathee [17].

Finally, we clarify the importance of the notion (E.A.) property in a convex metric space in comparison to the hypothesis that the range set of one map is contained in the range set of another map.

**Definition 1.** *[10] Let $(S, \rho)$ be a metric space. A continuous mapping $W : S \times S \times [0, 1] \to S$ is called a convex structure on S, if for all $x, y \in S$ and $\lambda \in [0, 1]$, we have*

$$d(u, W(x, y, \lambda)) \leq \lambda d(u, x) + (1 - \lambda)d(u, y)$$

*for all $u \in S$. A metric space $(S, \rho)$ equipped with a convex structure is called a convex metric space.*

Let $M$ be a subset of a convex metric space $(S, \rho)$. The set $M$ is said to be

(i)    convex if $W(x, y, \lambda) \in M$ for all $x, y \in M$ and $\lambda \in [0, 1]$;

(ii)    $q$-starshaped if there exists $q \in M$ such that $W(x, q, \lambda) \in M$ for all $x \in M$ and $\lambda \in [0, 1]$.

In addition, the map $I : M \to M$ is said to be

(i)    affine if $M$ is convex and $I(W(x, y, \lambda)) = W(Ix, Iy, \lambda)$ for all $x, y \in M$ and $\lambda \in [0, 1]$;

(ii)    $q$-affine if $M$ is $q$-starshaped and $I(W(x, q, \lambda)) = W(Ix, q, \lambda)$ for all $x \in M$ and $\lambda \in [0, 1]$.

Clearly, each convex set $M$ is $q$-starshaped for any $q \in M$ but the converse assertion is not necessarily true (see Example 7 of [19]).

**Definition 2.** *[10] A convex metric space $(S, \rho)$ is said to satisfy the Property (I), if for all $x, y, z \in S$ and $\lambda \in [0, 1]$, we have $\rho(W(x, z, \lambda), W(y, z, \lambda)) \leq \lambda \rho(x, y)$.*

Notice that Property (I) is always satisfied in a normed linear space and each of its convex subsets.

**Definition 3.** *[19] Let $T, I : S \to S$ be mappings on a metric space $(S, \rho)$. The pair $(T, I)$ is said to be compatible if*

$$\rho(TIx_n, ITx_n) \to 0$$

*whenever $\{x_n\}$ is a sequence in S such that*

$$Tx_n, \ Ix_n \to t \in S$$

**Definition 4.** *[16] Let $T, I : S \to S$ be mappings on a metric space $(S, \rho)$. The pair $(T, I)$ is said to satisfy (E.A.) property if there is a sequence $\{x_n\} \in S$ such that*

$$Tx_n, \ Ix_n \to t \in S$$

**Definition 5.** *Let $(S, \rho)$ be a metric space and $T, I : S \to S$. Then the pair $(T, I)$ is said to be reciprocally continuous if*

$$\lim_{n \to +\infty} TIx_n = Tt \quad and \quad \lim_{n \to +\infty} ITx_n = It$$

*whenever $\{x_n\}$ is a sequence in S such that $\lim_{n \to +\infty} Tx_n = \lim_{n \to +\infty} Ix_n = t$ for some $t \in X$.*

It is easy to see that if $T$ and $I$ are continuous, then the pair $(T, I)$ is reciprocally continuous but the converse is not true in general (see Example 2.3 of [20]).

Moreover, in the setting of common fixed point theorems for compatible pairs of self-mappings satisfying some contractive conditions, continuity of one of the mappings implies their reciprocal continuity.

**Definition 6.** *A pair $(T, I)$ of self-maps of a metric space $(S, \rho)$ is said to be sub-compatible if there exists a sequence $\{x_n\}$ such that*

$$\lim_{n \to +\infty} Tx_n = \lim_{n \to +\infty} Ix_n = t \ \text{ for some } \ t \in X \ \text{ and } \ \lim_{n \to +\infty} \rho(TIx_n, ITx_n) = 0.$$

Recently, Rouzkard et al. [18] proved the following common fixed point theorem for the pair of compatible maps in a convex metric space.

**Theorem 1.** *Let C be a nonempty closed convex subset of a convex metric space $(X, \rho)$ satisfying the Property $(I)$. Denote $[x, q] = \{W(x, q, k) : 0 \le k \le 1\}$ where W is a convex structure on the metric space.*

If $T$ and $I$ are compatible self-maps defined on $C$ such that $I(C) = C$, $I$ is $q$-affine and nonexpansive, which satisfy the inequality

$$\rho(Tx, Ty) \le \rho(Ix, Iy) + \frac{(1-k)}{k} \max\{\rho(Ix, [Tx, q]), \rho(Iy, [Ty, q])\} \tag{1}$$

for all $x, y \in C$, where $1/2 < k < 1$, then $T$ and $I$ have a common fixed point provided $cl(T(C))$ is compact and $T$ is continuous.

### 3. Results
*3.1. Compatibility in Proof of Theorem 1.*

Let us recall the lines of the proof given in Rouzkard et al. [18]. First of all, for each $n \in \mathbb{N}$, the authors define $T_n : C \to C$ by

$$T_n x = W(Tx, q, k_n) \ \text{ for all } \ x \in C, \tag{2}$$

where $k_n$ is a sequence in $(\frac{1}{2}, 1)$ such that $k_n \to 1$. Afterward, to accomplish the compatibility of the maps $T_n$ and $I$ for each $n \in \mathbb{N}$, the authors choose an arbitrary sequence $\{x_m\}$ in $C$ such that

$$\lim_{m \to +\infty} Ix_m = \lim_{m \to +\infty} T_n x_m = t \in C \tag{3}$$

Using the definition of $T_n$, it has been written that

$$\begin{aligned}
\rho(Tx_m, T_n x_m) &= \rho(Tx_m, W(Tx_m, q, k_n)) \\
&\le k_n \rho(Tx_m, Tx_m) + (1 - k_n)\rho(Tx_m, q) \\
&= (1 - k_n)\rho(Tx_m, q).
\end{aligned}$$

Then by taking $m \to +\infty$ and using (3), the authors get

$$\rho(\lim_{m \to +\infty} Tx_m, t) \le (1 - k_n)\rho(\lim_{m \to +\infty} Tx_m, q). \tag{4}$$

Again, on making $n \to +\infty$ in (4), the authors wrote the following (see [18], page 323, line 20–21)

$$\rho(\lim_{m \to +\infty} Tx_m, t) \le 0. \tag{5}$$

Then, by using this expression, the authors claim the compatibility of the maps $T_n$ and $I$ for each $n \in \mathbb{N}$.

Here, it is pertinent to mention that the compatibility of the maps $T_n$ and $I$ is to be shown for each $n \in \mathbb{N}$ and so the compatibility of $T_n$ and $I$ is to be shown for arbitrarily fixed natural number $n$. If $n$ is fixed, then it is superfluous to approach $n \to +\infty$, therefore (5) is not valid because this is obtained by taking $n \to \infty$ in (4). So the compatibility of the maps $T_n$ and $I$ for each $n \in \mathbb{N}$ proved by this way is totally wrong. The same mistake

occurred when the authors tried to prove the reciprocal continuity of $T_n$ and $I$ for each $n \in N$ (see [18], page 324, line 3–15).

*3.2. Modified Version of Theorem 1*

The following definition given by Kumar and Rathee [17] is required to prove the modified version of Theorem 1.

**Definition 7.** *Let M be a q-starshaped subset of a convex metric space $(S, \rho)$ and let $T, I : M \to M$ with $q \in F(I)$. The pair $(T, I)$ is said to satisfy (E.A.) property with respect to q if there exists a sequence $\{x_n\}$ in M such that for all $\lambda \in [0, 1]$.*

$$\lim_{n \to +\infty} Ix_n = \lim_{n \to +\infty} T_\lambda x_n = t \ \text{ for some } \ t \in M, \tag{6}$$

*where $T_\lambda x = W(Tx, q, \lambda)$.*

The following lemma is a direct consequence of Theorem 3.2 of Rouzkard et al. [18].

**Lemma 1.** *Let T and I be self-maps of a metric space $(S, \rho)$. If the pair $(T, I)$ is sub-compatible, reciprocally continuous and satisfies the inequality*

$$\rho(Tx, Ty) \leq a\, \rho(Ix, Iy) + (1 - a) \max\{\rho(Ix, Tx), \rho(Iy, Ty)\} \tag{7}$$

*for all $x, y \in X$, where $0 < \alpha < 1$. Then T and I have a unique common fixed point in X.*

Now we modify Theorem 1 by replacing the condition $I(M) = M \supseteq T(M)$ with the assumption that the pair $(T, I)$ satisfies (E.A.) property with respect to some $q \in M$.

**Theorem 2.** *Let M be a nonempty q-starshaped subset of a convex metric space $(X, \rho)$ with Property (I) and let T and I be continuous self-maps of M such that the pair $(T, I)$ satisfies (E.A.) property with respect to q. Assume that I is q-affine, $cl(T(M))$ is compact. If T and I are compatible and satisfy the inequality*

$$\rho(Tx, Ty) \leq \rho(Ix, Iy) + \frac{1 - k}{k} \max\{\rho(Ix, [Tx, q]), \rho(Iy, [Ty, q])\} \tag{8}$$

*for all $x, y \in M$, where $\frac{1}{2} < k < 1$, then T and I have a common fixed point in M.*

**Proof.** For each $n \in \mathbb{N}$, we define $T_n : M \to M$ by

$$T_n(x) = W(Tx, q, k_n) \ \text{ for all } \ x \in M, \tag{9}$$

where $k_n$ is a sequence in $(\frac{1}{2}, 1)$ such that $k_n \to 1$.

Now, we have to show that for each $n \in \mathbb{N}$, the pair $(T_n, I)$ is sub-compatible. Since $T$ and $I$ satisfy (E.A.) property with respect to $q$, there exists a sequence $\{x_m\}$ in $M$ such that for all $\lambda \in [0, 1]$

$$\lim_{m \to +\infty} Ix_m = \lim_{m \to +\infty} T_\lambda x_m = t \in M, \tag{10}$$

where $T_\lambda x_m = W(Tx_m, q, \lambda)$.

Since $k_n \in (0, 1)$, in light of (9) and (10), for each $n \in N$, we have

$$\begin{aligned}
\lim_{m \to +\infty} T_n x_m &= \lim_{m \to +\infty} W(Tx_m, q, k_n) \\
&= \lim_{m \to \infty} T_{k_n} x_m = t \in M.
\end{aligned}$$

Thus, we have

$$\lim_{m \to +\infty} Ix_m = \lim_{m \to +\infty} T_n x_m = t \in M. \tag{11}$$

Now using the fact that $I$ is $q$-affine and Property (I) is satisfied, we get

$$
\begin{aligned}
\rho(T_n Ix_m, IT_n x_m) &= \rho(W(TIx_m, q, k_n), I(W(Tx_m, q, k_n))) \\
&= \rho(W(TIx_m, q, k_n), W(ITx_m, q, k_n)) \\
&\leq k_n \, \rho(TIx_m, ITx_m). \tag{12}
\end{aligned}
$$

Since $(T, I)$ satisfies (E.A.) property with $T$ and $I$ are compatible, in view of (10) we have

$$\lim_{m \to +\infty} \rho(TIx_m, ITx_m) = 0.$$

Now, letting $m \to \infty$ in (12), we obtain

$$\lim_{m \to +\infty} \rho(T_n Ix_m, IT_n x_m) = 0. \tag{13}$$

Hence, on account of (11) and (13), it follows that the pair $(T_n, I)$ is sub-compatible for each $n \in \mathbb{N}$. Since $T$ and $I$ are continuous, for each $n \in \mathbb{N}$, the pair $(T_n, I)$ is reciprocally continuous. Furthermore, by (8),

$$
\begin{aligned}
\rho(T_n x, T_n y) &= \rho(W(Tx, q, k_n), W(Ty, q, k_n)) \\
&\leq k_n \, \rho(Tx, Ty) \\
&\leq k_n [\rho(Ix, Iy) + \frac{1 - k_n}{k_n} \max\{dist(Ix, [Tx, q]), dist(Iy, [Ty, q])\}] \\
&\leq k_n \, \rho(Ix, Iy) + (1 - k_n) \max\{\rho(Ix, T_n x), \rho(Iy, T_n y)\} \tag{14}
\end{aligned}
$$

for each $x, y \in M$ and $\frac{1}{2} < k_n < 1$. By Lemma 1, for each $n \in \mathbb{N}$, there exists $x_n \in M$ such that $x_n = Ix_n = T_n x_n$.

Now the compactness of $cl(T(M))$ implies that there exists a sub-sequence $\{Tx_m\}$ of $\{Tx_n\}$ such that $Tx_m \to z$ as $m \to +\infty$. Further, it follows that

$$x_m = T_m x_m = W(Tx_m, q, k_m) \to z \quad \text{as} \quad m \to +\infty.$$

Then, by the continuity of $T$ and $I$, we obtain $Iz = z = Tz$ and so $z$ is a common fixed point of $T$ and $I$. □

The following remark clarifies that in the context of a convex metric space, the notion of (E.A.) property introduced by Kumar and Rathee [17] for proving the common fixed point theorems has importance in comparison to the hypothesis that a range set of one map is contained in the range set of another map.

**Remark 1.**

(a) *In 2011, Haghi et al. [21] showed that several common fixed point generalizations in the theory of fixed point are not a real generalization because they can be obtained from the corresponding fixed point theorems. After the critical analysis of this paper, we reached the conclusion that the claim of Haghi et al. [21] is true only in the case if we make the assumption that the range set of one map is contained in the range set of another map.*
*So, keeping this in view, we replaced the condition $I(M) = M \supseteq T(M)$ of Theorem 1 with the assumption that the pair $(T, I)$ satisfies (E.A.) property with respect to some $q \in M$ and due to this we have been able to obtain the modified and correct version of Theorem 1 in the form of Theorem 2.*

(b)  (see Example 17 of [16]) Let $S = \mathbb{R}$ with usual metric and $M = [0, 1]$. Define $T, I : M \to M$ by

$$T(x) = \begin{cases} \frac{1}{2} & \text{if } 0 \le x \le \frac{1}{2} \\ \frac{x}{2} + \frac{1}{4} & \text{if } \frac{1}{2} \le x \le 1. \end{cases} \quad \text{and} \quad I(x) = \begin{cases} \frac{1}{2} & \text{if } 0 \le x \le \frac{1}{2} \\ 1 - x & \text{if } \frac{1}{2} \le x \le 1. \end{cases}$$

*Then $(S, \rho)$ is a convex metric space with $W(x, y, \lambda) = \lambda x + (1 - \lambda)y$. It is easy to verify that the pair $(T, I)$ satisfies (E.A.) property with respect to $q = \frac{1}{2}$, but the pair violates the condition that the range set of one map is contained in the range set of another map since $T(M) = [\frac{1}{2}, \frac{3}{4}] \not\subseteq [0, \frac{1}{2}] = I(M)$ and $I(M) = [0, \frac{1}{2}] \not\subseteq [\frac{1}{2}, \frac{3}{4}] = T(M)$.*

In this way, we can say that there are certain pairs of self-maps, namely $T$ and $I$, defined on a set (say $M$), which satisfies (E.A.) property in the set up of convex metric space but violates the condition $T(M) \subseteq I(M)$. Thus, the common fixed point theorems in which the pair of maps satisfy (E.A.) property with some other hypotheses will ensure the existence of a common fixed point for such maps.

**Remark 2.** *As an application of Theorem 1, the authors in [18] obtained two more theorems (see Theorems 4.1 and 4.2 of [18]). Since we have quoted a gap in the proof of Theorem 1, Theorems 4.1 and 4.2 of [18] are no longer valid. Thus, these theorems can also be modified by using the notion of (E.A.) property in the set up of a convex metric space.*

## 4. Conclusions

In this work, a gap in the proof of the main result of Rouzkard et al. (The Bulletin of the Belgian Mathematical Society 2012) is detected. Then after, utilizing the concept of (E.A.) property in convex metric space, we obtained an alternative and correct version of this result.

In the set up of a convex metric space, the notion of (E.A.) property introduced by Kumar and Rathee [17] for proving the common fixed point theorems is more important than the hypothesis that a range set of one map is contained in the range set of another map and it develops some new extensions.

**Author Contributions:** Authors contributed equally to this work. All authors have read and agreed to the published version of the manuscript.

**Funding:** This research received no external funding.

**Institutional Review Board Statement:** Not applicable.

**Informed Consent Statement:** Not applicable.

**Data Availability Statement:** This research did not report any data.

**Acknowledgments:** Thanks to Nasser Shahzad, King Abdulaziz University, Saudi Arabia, for providing us the preprint copy of Haghi et al. [21].

**Conflicts of Interest:** The authors declare no conflict of interest.

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
