# Peer review of "Note on Common Fixed Point Theorems in Convex Metric Spaces"

_axioms, doi:10.3390/axioms10010028_

Round 1
Reviewer 1 Report
To a certain extent, I think the manuscript titled "A CRUCIAL NOTE ON FIXED POINT THEOREMS" is interesting. In virtue of (EA) property, the authors modify the proof of Theorem 1 (Rouzkard et al, The Bulletin of the Belgian Mathematical Society 2012). However, I think the proof process of Theorem 2 in this manuscript is too simple because of the strong condition "(EA) property". Can this condition be weakened?In addition, I notice that the range of k in (1) is 1/2 to 1, can the range of k be extended to the interval [0,1]?Besides,I suggest that the grammar and preciseness of the article should be strengthened.
Reviewer 2 Report
In this paper, the authors draw attention to the gap made in the paper [13] ("New common fixed point theorems and invariant
approximation in convex metric spaces ", Bull. Belg. Math. Soc. Simon Stevin 19 (2012), 311–328) which deals with the problem of common fixed point within convex metric spaces.
Authors correct the theorem with the additional assumption that the pair $ (T, I) $ satisfies (E.A.) property with respect to some q, instead of the assumption given in [13].
For the easier of monitoring, I suggest that the authors add a definition of compatible mappings, as well as define [x, y] (eq. (1)). Whether in Definition 4 should stay iff instead of if?
I would also ask the authors to explain in more detail why from eq. (12) follows $$ \lim_{m \to \infty} \rho(TIk_m, ITk_m) = 0.$$
After these changes and explanations, I suggest the paper "A CRUCIAL NOTE ON FIXED POINT THEOREMS" be accept for publication in the journal Axioms.

Reviewer 3 Report
The paper under review presents a gap in the proof of a published result. Moreover, under suitable hypotheses, the authors correct the previous theorem.
However, the current paper has some deficiencies. In the Introduction section is formulated a Theorem, but this fact is not recommended; in the formulation of this theorem, the authors use a notion which is not defined previously (it appears in Section 2). The recommendation is to present this theorem in Section 2. Also, re-name Section 4 as Conclusions and do not use the same text as in Abstract or Introduction.
Reviewer 4 Report
All is as in the report

Round 2
Reviewer 1 Report
I think the revised manuscript is suitable for publication in this journal.
Author Response
REVIEWER 1: Comments and Suggestions for Authors
To a certain extent, I think the manuscript titled "A CRUCIAL NOTE ON FIXED POINT THEOREMS" is interesting. In virtue of (EA) property, the authors modify the proof of Theorem 1 (Rouzkard et al, The Bulletin of the Belgian Mathematical Society 2012). However, I think the proof process of Theorem 2 in this manuscript is too simple because of the strong condition "(EA) property". Can this condition be weakened?In addition, I notice that the range of k in (1) is 1/2 to 1, can the range of k be extended to the interval [0,1]?Besides,I suggest that the grammar and preciseness of the article should be strengthened.
Response:
Point 1. The condition of satisfying (EA) Property is essential in this proof. However it would be appropriate to examine the problem of replacing the (EA) property with a weaker condition in further research. There is no such an example in the literature so on. Here we just modified the proof of Theorem 1 (Rouzkard et al, The Bulletin of the Belgian Mathematical Society 2012) with the specified conditions. Thanks to the reviewer for valuable suggestions on a possible future research.
Point 2. The range of k in Equation (1) is 1/2 to 1 as in the original assumptions of [Rouzkard et al; Bull. Belg. Math. Soc. Simon Stevin 19 (2012), 311–328]. In the proving techniques of Theorems 1 and 2, the range of k can not be extended to the interval [0,1]. But in general for the range containing also (0, ½) is an open question.